

# Attention enhanced capsule network for text classification by encoding syntactic dependency trees with graph convolutional neural network

Xudong Jia and Li Wang

College of Data Science, Taiyuan University of Technology, Taiyuan, Shanxi, China

## ABSTRACT

Text classification is a fundamental task in many applications such as topic labeling, sentiment analysis, and spam detection. The text syntactic relationship and word sequence are important and useful for text classification. How to model and incorporate them to improve performance is one key challenge. Inspired by human behavior in understanding text. In this paper, we combine the syntactic relationship, sequence structure, and semantics for text representation, and propose an attention-enhanced capsule network-based text classification model. Specifically, we use graph convolutional neural networks to encode syntactic dependency trees, build multi-head attention to encode dependencies relationship in text sequence, merge with semantic information by capsule network at last. Extensive experiments on five datasets demonstrate that our approach can effectively improve the performance of text classification compared with state-of-the-art methods. The result also shows capsule network, graph convolutional neural network, and multi-headed attention has integration effects on text classification tasks.

## INTRODUCTION

Text classification is the basic task of text analysis with broad applications in topic labeling, sentiment analysis, and spam detection. Text representation is one important way for classification. From the analysis of text structure, the text is a sequence of words by certain rules. Sequences of different word orders show different meanings. The sequence structure of a text contains important information about the semantics of the text. Moreover, words in a text are contextual, the distance dependence between words in the sequence structure will also affect the meaning of the text. From the analysis of text composition, the text is composed of words or phrases with different syntactic functions according to certain syntactic relations. For humans, syntactic relations are the basis for editing and reading texts. According to the syntactic relation, we can understand the subject, predicate, and object of the text to promptly understand the semantics of the text. Whether it is text representation or text classification by text representation, it is the key

Corresponding authors
Xudong Jia, jxd1005@163.com
Li Wang, wangli@tyut.edu.cn

and important question for our research to extract information from the sequence structure and syntactic relations of text.

Traditional methods represent text with handcrafted features, such as bag-of-words (*Joachims, 1998*; *Mccallum & Nigam, 1998*), N-grams (*Lin & Hovy, 2003*), and TF-IDF (*Zhang, Yoshida & Tang, 2008*), and then adopted machine learning algorithms such as Naive Bayes (*Mccallum & Nigam, 1998*), logistic regression (*Genkin, Lewis & Madigan, 2007*), support vector machine (*Joachims, 1998*), *etc.* for classification. These methods ignore the word order and semantic information, which has an important role in understanding the semantics of the text (*Pang, Lee & Vaithyanathan, 2002*). On the contrary, other methods use word2vec and glove (*Mikolov et al., 2013*; *Pennington, Socher & Manning, 2014*) to represent text that can represent the semantic information of words, and then adopted convolutional neural network (CNN) (*Kim, 2014*; *Zhang, Zhao & Lecun, 2015*; *Conneau et al., 2017*), long short-term memory networks (LSTM) (*Mousa & Schuller, 2017*), Capsule networks (CapsNet) (*Zhao et al., 2018*) and other deep neural networks for text classification. These methods can effectively encode the word order and semantic information. *Kim (2014)* proposed a CNN model that adopted convolution filters to extract local text semantic features for text classification, leading to the model loss of a part of location information (*Zhao et al., 2018*). Capsule networks with vector neural units and dynamic routing algorithms can effectively overcome the disadvantages of CNN (*Sabour, Frosst & Hinton, 2017*; *Xi, Bing & Jin, 2017*). *Zhao et al. (2018)* optimized the capsule networks for text classification (Capsule-A), and further experiments show the effectiveness of capsule networks for text classification. The characteristics of capsule networks are our starting point. However, this model has limitations in recognizing text with semantic transitions since it cannot encode long-distance dependencies and the global topology, which affects the effectiveness of complex text classification. The attention-based approach is effective in overcoming the problem of distance dependencies of sequence structure. Both Transformer (*Vaswani et al., 2017*) and Bidirectional Encoder Representations from Transformers (BERT) (*Alaparthi & Mishra, 2020*) use multi-head attention as a basic unit to extract text features. It's part of our motivation that it can extract the distance dependencies information of the text from different subspaces.

Syntactic information is a different attribute for text. It describes the syntactic dependency relationship between words in a sentence. It can be represented as the syntactic dependency tree and show the global topology. Figure 1 is a syntactic dependency tree, where 'monkey' is the subject of the predicate 'eats', and 'apple' is its object. GCN is a graph convolutional network that operates on graphs and induces embeddings of nodes based on the properties of their neighborhoods. It can capture the information of the immediate neighbors of nodes at most K hops away (*Marcheggiani & Titov, 2017*; *Duvenaud et al., 2015*). *Marcheggiani & Titov (2017)* used GCN to encode syntactic dependency trees to generate word representations, and combined with long short-term memory networks for semantic role labeling tasks. These works show that GCN can effectively extract syntactic information in syntactic dependency trees. This is another motivation for us.

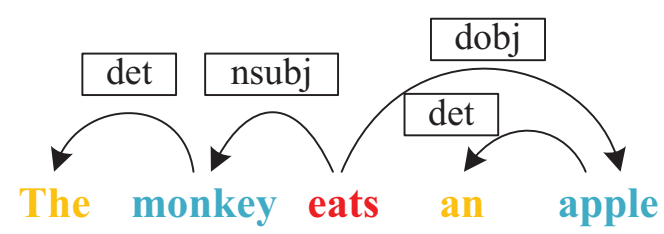

**Figure 1 The example of syntactic dependency tree, where 'monkey' is the subject of the predicate 'eats', and 'apple' is its object.**

The text syntactic relationship and word sequence are important and useful for text classification. How to model and incorporate them to improve performance is one key challenge. In this paper, we combine the syntactic relationship, sequence structure, and semantics for text representation, and propose a novel model that utilizes GCN for syntactic relationship, multi-head attention for words, and corporate them in capsule network for text classification.

The contributions of this paper can be summarized as follows:

- We incorporate syntactic relationship, sequence structure, and semantics for text representation.
- We introduce GCN to extract syntactic information for dependencies relationship representation.
- We build multi-head attention to encode the different influences of words to enhance the effect of capsule networks on text classification.
- We show that CapsNet, GCN, and multi-head attention have an integration effect for text classification.

# RELATED WORK

## Machine learning-based methods

Early methods adopted the typical features such as bag-of-words (*Joachims, 1998*; *Mccallum & Nigam, 1998*), N-grams (*Lin & Hovy, 2003*), and TF-IDF (*Zhang, Yoshida & Tang, 2008*) features as input and utilized machine learning algorithms such as support vector achine (SVM) (*Joachims, 1998*), logistic regression (*Genkin, Lewis & Madigan, 2007*), naive Bayes (NB) (*Mccallum & Nigam, 1998*) for classification. However, these methods ignore the text word order and semantic information, and usually heavily rely on laborious feature engineering.

## Deep learning-based methods

With the introduction of distributed word vector representation (Word Embedding) (*Mikolov et al., 2013*; *Pennington, Socher & Manning, 2014*), neural networks-based methods have substantially improved the performance of text classification tasks by encoding text semantics.

CNN was first applied to image processing. *Kim (2014)* proposed the CNN-based text classification model (TextCNN). The model uses convolution filters to extract local semantic features and improved upon the state of the art on four out of seven tasks. *Zhang, Zhao & Lecun (2015)* proposed the character-level CNN model, which extracts semantic

information from character-level original signals for text classification tasks. *Conneau et al. (2017)* proposed very deep convolutional networks to learn the hierarchical representation for text classification. Being a spatially sensitive model, CNN pays a price for the inefficiency of replicating feature detectors on a grid.

Recently, *Sabour, Frosst & Hinton (2017)* proposed the CapsNet model, which uses vector neural units and dynamic routing update mechanisms, and verified its superiority in image classification. *Zhao et al. (2018)* proposed the text classification model based on CapsNet (Capsule-A), which adopted CapsNet to encode text semantics, and proved that its classification effect is superior to CNN and LSTM. In CapsNet, the feature is represented by a capsule vector instead of a scalar (activation value output by neuron). Different dimensions in a vector can represent different properties of a feature. For a text feature, it often means different meanings in different semantic relations. We use capsules to represent text features to learn the semantic information of different dimensions of text features. On the other hand, the similarity between features at different levels is different. For building a high-level feature, lower levels with high similarity have a higher weight. CapsNet can learn this similarity relationship through a dynamic routing algorithm. Although CapsNet can effectively improve coding efficiency, it still has limitations in recognizing text with semantic transitions.

Attention mechanisms are widely used in tasks such as machine translation (*Vaswani et al., 2017*) and speech recognition (*Chorowski et al., 2014*). *Lin et al. (2017)* proposed the self-attention mechanism that can encode long-range dependencies. *Vaswani et al. (2017)* proposed a machine translation model (Transformer) based on multi-head attention. *Alaparthi & Mishra (2020)* proposed a pre-trained model of language representation (BERT) that also takes multi-head attention as its basic component. The basic unit of multi-head attention is scaled dot-product attention. Multi-head attention allows the model to jointly attend to information from different representation subspaces at different positions. Attention is to extract the long-distance dependencies in the text by calculating the similarities between words in the text. The words in the text can express different meanings in different semantic scenarios. The representation of words is different in different semantic spaces. Similar to ensemble learning, multi-head attention can put text in different semantic spaces to calculate attention and get integrated attention. The location information cannot be obtained by relying solely on the attention mechanism, and location information also has an important influence on understanding text semantics. *Kim, Lee & Jung (2018)* proposed a text sentiment classification model combining attention and CNN, but it is still limited by the disadvantages of CNN. Although BERT has been particularly effective on many tasks, it requires a lot of data and computing resources for pre-training. Therefore, our research is still valuable.

## Syntactic information-based methods

CNN, RNN, and most deep learning-based methods always utilized word local topology to represent text. Word order and semantic, text syntactic information all have important influences on text classification. Some researchers have done some work on text syntactic information for different tasks. A text is made up of words that represent different

syntactic elements, such as subject, predicate, object, and so on. Different syntactic elements are interdependent. A syntax dependency tree is a kind of tree structure, which describes the dependency relationship between words. Figure 1 shows an example of a syntax dependency tree. There are many tools (like StanfordNLP) to generate syntactic dependency trees by analyzing syntactic dependency relations.

*Eriguchi, Tsuruoka & Cho (2017)* adopted RNN to integrate syntactic information for machine translation. *Le & Zuidema (2014)* introduced RNN to model syntax dependency tree. *Eriguchi, Tsuruoka & Cho (2017)* used sequential LSTM and tree-LSTM to extract syntactic relations.

The syntactic dependency tree is also a kind of graph data. *Bastings et al. (2017)* used GCN to encode the syntactic dependency tree and combined it with the CNN for machine translation. *Marcheggiani & Titov (2017)* used GCN to encode syntactic dependency trees to generate word representations and combined it with LSTM for the role labeling task. These works show that GCN can effectively extract syntactic information in syntactic dependency trees. The syntactic relationship is presented as a tree structure. A tree is also a form of a graph. Secondly, the sequence structure-based model represents the syntactic tree as a sequence according to some rules of nodes. But the sequence is directional, and nodes in a tree are not sequential. GCN can directly consider the relationship between nodes in a syntax tree.

In summary, we aim to propose a novel model named Syntax-AT-CapsNet that uses multi-head attention to extract long-distance dependencies information and that uses GCN to encode syntactic dependency trees to extract syntactic information, which enhances the effect of capsule networks on text classification tasks.

# SYNTAX-AT-CAPSNET MODEL

Our Syntax-AT-CapsNet model consists of the following three modules as depicted in (Fig. 2).

- Attention module. It is composed of an attention layer that adopts multi-head attention. It encodes the dependency relationship between words in the text sequence and important word information to form a text representation.
- Syntax module. It is composed of GCN. It encodes the syntax dependency tree, extracts the syntactic information in the text to for a text representation.
- Capsule network module. It is a capsule network with 5-layer. Based on text representations output by the Attention module and the Syntax module, it further extracts text semantic and structural information to classify the text.

## Input

The input of the Syntax-AT-CapsNet model is defined as the sentence matrix $X$:

$$X = [x_1, x_2, \ldots, x_L] \in \mathbb{R}^{L \times d}, \tag{1}$$

where $x_i \in \mathbb{R}^d$ is the word vector of the $i$-th word in the sentence, $L$ is the length of the sentence, and $d$ is the embedding vector size of words.

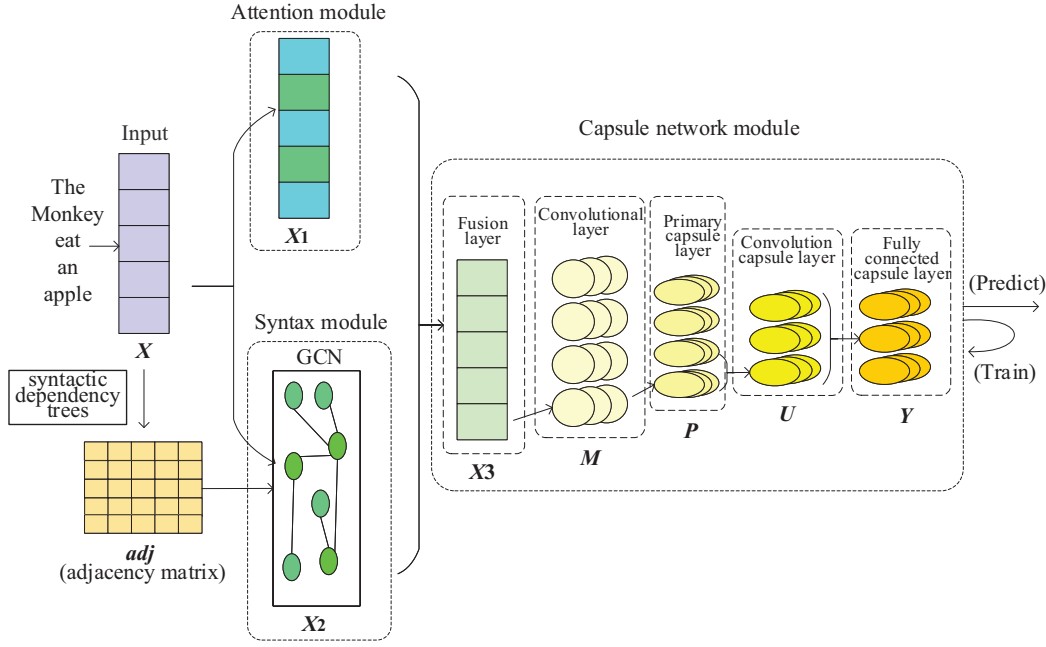

**Figure 2** The architecture of the Syntax-AT-CapsNet model, which consists of three modules: the attention module, the syntax module and the capsule network module.

## Attention module

The Attention module is shown in (Fig. 3). The calculation of attention in this module can be divided into five steps.

First step, linearly transform the input sentence matrix $X$ and divide it into three matrices: $\mathbf{Q} \in \mathbb{R}^{L \times d}$, $\mathbf{K} \in \mathbb{R}^{L \times d}$, $\mathbf{V} \in \mathbb{R}^{L \times d}$:

$$[\boldsymbol{Q}, \boldsymbol{K}, \boldsymbol{V}] = \boldsymbol{spilt}[\boldsymbol{X} \cdot \boldsymbol{W}], \tag{2}$$

where $\boldsymbol{W} \in \mathbb{R}^{d \times 3d}$ is the transform matrix, and *spilt* denotes the division operation.

Second step, linearly projects the matrices $\boldsymbol{Q}, \boldsymbol{K}, \boldsymbol{V}$ onto $h$ different linear subspaces:

$$
\begin{aligned}
[\boldsymbol{Q_1}, \dots, \boldsymbol{Qh}] &= [\boldsymbol{Q} \cdot \boldsymbol{W^{Q_1}}, \dots, \boldsymbol{Q} \cdot \boldsymbol{W^{Q_h}}], \\
[\boldsymbol{K_1}, \dots, \boldsymbol{Kh}] &= [\boldsymbol{K} \cdot \boldsymbol{W^{K_1}}, \dots, \boldsymbol{K} \cdot \boldsymbol{W^{K_h}}], \\
[\boldsymbol{V_1}, \dots, \boldsymbol{Vh}] &= [\boldsymbol{V} \cdot \boldsymbol{W^{V_1}}, \dots, \boldsymbol{V} \cdot \boldsymbol{W^{V_h}}],
\end{aligned}
\tag{3}
$$

where $\boldsymbol{Qi} \in \mathbb{R}^{L \times \frac{d}{h}}$, $\boldsymbol{Ki} \in \mathbb{R}^{L \times \frac{d}{h}}$, $\boldsymbol{Vi} \in \mathbb{R}^{L \times \frac{d}{h}}$ is the mapping of $\boldsymbol{Q}, \boldsymbol{K}, \boldsymbol{V}$ on the $i$-th

subspace, $\boldsymbol{W^{Qi}} \in \mathbb{R}^{d \times \frac{d}{h}}$, $\boldsymbol{W^{Ki}} \in \mathbb{R}^{d \times \frac{d}{h}}$, $\boldsymbol{W^{Vi}} \in \mathbb{R}^{d \times \frac{d}{h}}$ is the transform matrix,

$i = [1, \dots, h]$. The purpose of this step is to compute multiple attention values in parallel.

At the same time, the dimension of input matrix is reduced to reduce the calculation pressure caused by multiple calculation.

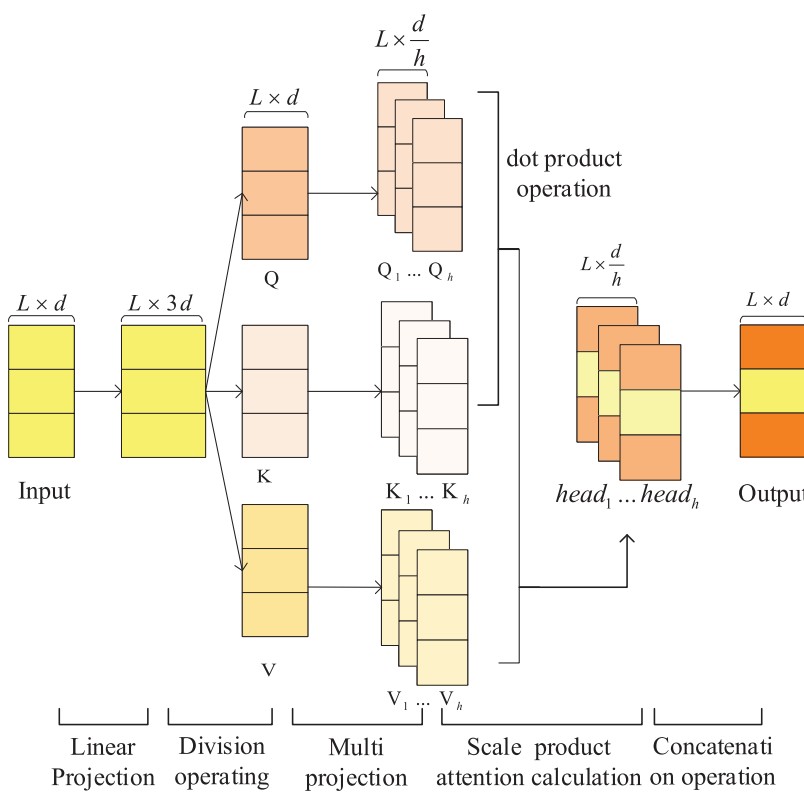

**Figure 3** The architecture of the attention module, which consists of three parallel attention computing units.

Third step, calculate the attention on each subspace in parallel:

$$head_i = softmax\left(Q_i \cdot K_i^T / \sqrt{d}\right) V_i, \tag{4}$$

where $head_i$ is the attention value on the $i$-th subspace, *softmax* denotes the *softmax* function (*Vaswani et al., 2017*). In fact, $Q_i$ and $K_i$ represents the sentence matrix on the subspace. It's divided by $\sqrt{d}$ in case the dot product gets too big. The weight of the sentence matrix on the subspace $V_i$ is obtained by calculating the dot product of $Q_i$ and $K_i$ and using a *softmax* function.

Fourth step, concat the attention values on each subspace and get the attention value of the entire sentence through linear transformation:

$$Multi\_head = concat(head_1, \ldots, head_h) W^M, \tag{5}$$

where $W^M \in \mathbb{R}^{d \times d}$ is the transform matrix, $Multi\_head$ is the attention value of the entire sentence, and *concat* denotes the concat operation.

The final step, Connect the attention value $Multi\_head$ to the original sentence matrix $X$ to get the sentence matrix output by the module:

$$X_1 = residual\_Connect(X, Multi\_head), \tag{6}$$

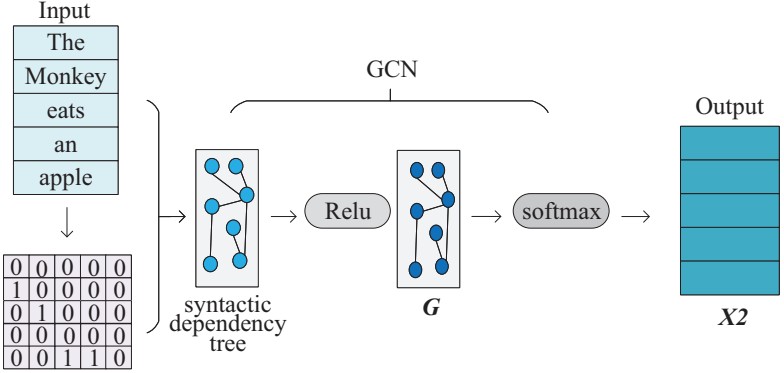

Adjacency matrix

Figure 4 The architecture of the syntax module, which includes the construction of the syntactic dependency tree and the GCN unit.

---

**Algorithm 1** Adjacency matrix construction algorithm.

**Input**: sentence matrix $X$

**Output**: adjacency matrix $adj$

1 define zero matrix $adj$

2 Use the StanfordNLP tool to generate the syntax dependency tree $Tree$;

3 **for** each edge <i, j> in $Tree$ **do**:

4     Put the element $adj[i,j]$ in the matrix $adj$ to 1;

5 **end for**;

6 **return** the adjacency matrix $adj$;

7 **end.**

---

where $X_1 \in \mathbb{R}^{L \times d}$ is the output of the attention module, and $residual\_Connect$ denotes the residual connection operation.

## Syntax module

The Syntax module is shown in (Fig. 4). The Syntax module uses GCN to encode syntactic dependency trees, which can encode syntactic relationships between words in a text into word vectors. The module first needs to use a natural language processing tool (we adopted StanfordNLP) to generate the syntactic dependency tree of the input sentence, and construct its adjacency matrix. The adjacency matrix construction algorithm in this paper is shown in Algorithm 1. Since the syntactic relationship between word nodes in the syntactic dependency tree has direction, when constructing the adjacency matrix, the syntactic dependency tree is used as a directed graph. In addition, in order not to be disturbed by its word vector, the node is not provided with a self-loop. As shown in (Fig. 5), the adjacency matrix corresponds to the example sentence "The Monkey eats an apple" shown in (Fig. 1) and is generated by our method. The input sentence matrix and

The Monkey eat an apple

| | The | Monkey | eat | an | apple |
|---|---|---|---|---|---|
| The | 0 | 0 | 0 | 0 | 0 |
| Monkey | 1 | 0 | 0 | 0 | 0 |
| eat | 0 | 1 | 0 | 0 | 0 |
| an | 0 | 0 | 0 | 0 | 0 |
| apple | 0 | 0 | 1 | 1 | 0 |

**Figure 5 The example of adjacency matrix for a syntactic dependent tree.**

adjacency matrix are further passed through the GCN to obtain a text representation containing syntactic information.

The calculation in the Syntax module can be divided into two steps.

First step, use StanfordNLP tool to generate syntactic dependency tree and construct adjacency matrix $adj$.

Second step, perform a two-layer graph convolution operation on the input sentence matrix $X$ and adjacency matrix $adj$:

$$G = relu((X \cdot adj)W^{t1}), \tag{7}$$
$$X_2 = softmax((G \cdot adj)W^{t2}), \tag{8}$$

where $G \in \mathbb{R}^{L \times d}$ is the output of the first layer graph convolution operation, $X_2 \in \mathbb{R}^{L \times d}$ is the output of the second layer graph convolution operation, and $adj \in \mathbb{R}^{L \times L}$ is adjacent matrix, $W^{t1}$ and $W^{t2} \in \mathbb{R}^{d \times d}$ are parameter matrices. In operation (7), the adjacency matrix and sentence matrix go through the first level graph-convolution operation. The relationship between nodes directly connected is obtained. Then, through operation (8), the relationship between nodes indirectly connected nodes is calculated.

## Capsule network module

The capsule network module is composed of a fusion layer, a convolution layer, a primary capsule layer, a convolution capsule layer, and a fully connected capsule layer. It uses the text representation output by the attention module and the syntax module as input to further extract features. Each layer in the module can extract different levels of features. By further combining low-level features to obtain higher-level features, and finally form a feature representation of the entire text for classification.

The first layer is the fusion layer, put the text representations $X_1$ and $X_2$ output by the syntax module and the attention module into a single layer network:

$$X_3 = W^{f1} \cdot X_1 + W^{f2}X_2, \tag{9}$$

where $W^{f1}, W^{f2} \in \mathbb{R}^L$. Two sentence matrices are combined by linear transformation through this step.

The second layer is the convolutional layer, which extracts N-gram phrase features at different positions in the text. This layer uses $k_1$ convolution filters to perform convolution on the sentence matrix $X_3$ to obtain the N-gram feature matrix $M$:

$$M = [M_1, \ M_2, \ldots, \ M_{k_1}] \in \mathbb{R}^{(L-N+1) \ \times \ k_1}, \tag{10}$$

where $M_{k_1} = [m_1, \ m_2, \ldots, \ m_{L-N+1}] \in \mathbb{R}^{L-N+1}$ is the $k_1$-th column vector in $M$, each element $m_i$ in this vector is obtained by operation (11):

$$m_i = f(W^{c1} \cdot x_{i:i+N-1} + b_1), \tag{11}$$

where $f$ denotes the nonlinear activation function, $W^{c1} \in \mathbb{R}^{N \times d}$ is the $k_1$-th convolution filter, $x_{i:i+N-1}$ denotes that $N$-word vectors in the sentence are connected in series, $b_1$ is bias item. The original features are extracted by convolution in this step.

The third layer is the primary capsule layer, which combines the N-gram phrase features extracted at the same location as capsules. This layer uses $k_2$ transformation matrices to transform the feature matrix $M$ into the primary capsule matrix $P$:

$$P = [P_1, \ P_2, \ldots, \ P_{k_2}] \in \mathbb{R}^{(L-N+1) \times k_2 \times l}, \tag{12}$$

where $P_{k_2} = [p_1, \ p_2, \ldots, \ p_{L-N+1}] \in \mathbb{R}^{(L-N+1) \times l}$ is the $k_2$-th column capsule in $P$, each capsule is obtained by operation (13):

$$p_i = g(W^{c2} \cdot M^i + b_2), \tag{13}$$

where $g$ denotes the nonlinear compression function, $W^{c2} \in \mathbb{R}^{k_2 \times 1 \times l}$ is the $k_1$th transformation matrix, $M^i$ is the $i$-th row vector in $M$, and $b_2$ is bias item. The primary capsules are constructed by linearly transforming the original features of the same location.

The fourth layer is the convolution capsule layer, which uses a shared transformation matrix to extract local capsules, similar to the convolution layer. This layer uses $k_3$ transformation matrices to perform capsule convolution operation on $P$ to obtain the capsule matrix $U$:

$$U = [U_1, \ U_2, \ldots, \ U_{k_3}] \in \mathbb{R}^{(L-N-N_1+2) \times k_3 \times l}, \tag{14}$$

where $U_{k_3} = [u_1, \ u_2, \ldots, \ u_{L-N-N_1+2}] \in \mathbb{R}^{(L-n_2+1) \times l}$ is the $k_3$-th column capsule in $U$, each capsule is obtained from the $N_1$ line capsules in P:

$$u_{L-N-N_1+2} \leftarrow P^{i:i+(N_1 \times k_2)-1}, \tag{15}$$

where $P^{i:i+(N_1 \times k_2)-1}$ represents $N_1$ rows of capsules, and each capsule is linearly converted to obtain a prediction vector:

$$\hat{p}_i = W^{c3} \cdot p_i + \hat{b}i, \tag{16}$$

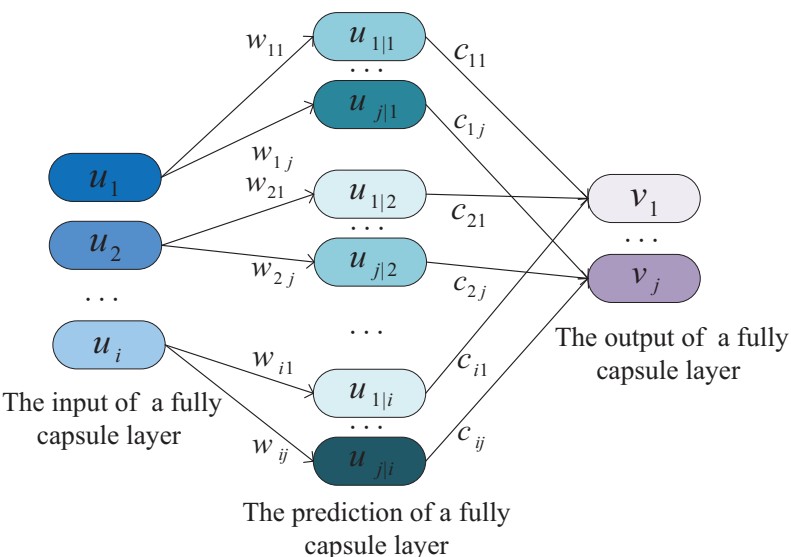

**Figure 6 The architecture of fully connected capsule layer, its input is the output of the previous capsule layer.**

where $W^{c3} \in \mathbb{R}^{l \times l}$ is the $k_3$-th conversion matrix, $\hat{b}_i$ is the offset term. It's the same thing as convolution, except the basic units become capsules. And the prediction vector is operated (17):

$$\boldsymbol{u}_r = g(\Sigma i \boldsymbol{c}_i \cdot \hat{\boldsymbol{p}} i), \tag{17}$$

where $g$ is a nonlinear activation function, $c_i$ is the coupling coefficient, which is updated with a dynamic routing algorithm (*Zhao et al., 2018*). The similarity between the primary capsules and the generated convolutional capsules is different within a window. A primary capsule with a high similarity should be given a higher weight. Equation (17) is based on this principle.

The last layer is the fully connected capsule layer, which is used to form the capsule Y representing the category:

$$\boldsymbol{Y} = \left[ \boldsymbol{y_1}, \ \boldsymbol{y_2}, \ldots, \ \boldsymbol{y_j} \right] \in \mathbb{R}^{j \times l}, \tag{18}$$

where $\boldsymbol{y_j} \in \mathbb{R}^l$ denotes the capsule of the $j$-th category. The capsules in $U$ are linearly transformed (16) to obtain the prediction vector $\boldsymbol{u}_{j|r}$, and the operation (17) is performed to obtain $y_j$. The fully connected capsule layer is shown in Fig. 6, it can also represent a convolution window operation at the fourth layer.

Finally, the modulus of the capsule vector representing the category in the fully connected capsule layer is taken as the probability of belonging to the category.

## Syntax-AT-CapsNet learning algorithm

The learning algorithm of Syntax-AT-CapsNet is shown in Algorithm 2. When the model is learning, the coupling coefficients are updated by the dynamic routing algorithm, and the global parameters of the model are updated by the back propagation algorithm.

| Algorithm 2 Syntax-AT-CapsNet learning algorithm. |
| --- |

**Input**: sentence matrix $X$; training batch Epoch;

**Output**: Trained Syntax-AT-CapsNet model parameters;

1 Initialize the parameters in the model;

2 **for** each training batch **do**:

    //Attention module: Step 3-Step 7

3      Linearly transform and divide the sentence matrix $X$ into $Q$, $K$, $V$;

4      Project $Q$, $K$, $V$ into h subspaces to obtain $Q_i$, $K_i$, $V_i$;

5      Calculate attention on each subspace to get $head_i$;

6      Concat each subspace $head_i$ and linearly transform to get $Multi\_head$;

7      Residuals connect $Multi\_head$ and $X$ as sentence matrix $X_1$;

    //Syntax module: Step 8-Step 9

8      Construct adjacency matrix $adj$

9      Perform graph convolution operation on $X$ and $adj$ to get sentence matrix $X_2$

    //Capsule network module: Step 10-Step 14

10      Put $X_1$ and $X_2$ into MLP get the sentence matrix $X_3$

11      Perform convolution operation on $X_3$ to obtain an N-gram matrix $M$;

12      Convert the N-gram feature matrix $M$ to the capsule matrix $P$;

13      Perform capsule convolution and routing on $P$ to obtain the capsule matrix $U$;

14      Perform capsule calculation and routing on $U$ to obtain category capsule $Y$;

15      Calculate loss and update parameters by back propagation;

16 **end for**;

17 **return** the trained Syntax-AT-CapsNet model parameters;

18 **end**.

Then the trained Syntax-AT-CapsNet model parameters can be obtained. During prediction, the classification results can be obtained through the sequential calculation of each module in the model.

# EXPERIMENTAL DETAILS

We did extensive experiments to verify the effect of Syntax-AT-CapsNet model on the single label and multi-label text classification tasks and designed more ablation experiments to demonstrate the role of each module.

## Data sets

We choose the following four datasets in our experiments: movie reviews (MR) (*Miwa & Bansal, 2016*), subjectivity dataset (Subj) (*Pang & Lee, 2004*), customer review (CR) (*Hu & Liu, 2004*), Reuters-21578 (*Lewis, 1992*).

    The data in MR, Subj, and CR have two categories and are used for the single-label classification tasks, where MR and Subj are composed of movie sentiment review data, CR is composed of product reviews from Amazon and Cnet. The Reuters-21578 test set

**Table 1 Data description: the number of training sets, validation sets and test sets of five kinds of data sets, the number of categories of each data set, and whether the data set belongs to single-label data or multi-label data.**

| Data set | Train/Vec/Test | Num of category | Test label |
|---|---|---|---|
| MR | 8632/960/1066 | 2 | Single |
| CR | 3053/340/377 | 2 | Single |
| Subj | 8099/900/1000 | 2 | Multi |
| Reuters-Full | 5000/500/2500 | 8 | Single/Multi |
| Reuters-Multi | 5000/500/300 | 8 | Multi |

consists of Reuters news documents. We selected 10,788 news documents under the 8 category labels related to economic and financial topics in Reuters-21578 and further divided them into two sub-datasets (Reuters-Full and Reuters-Multi). In Reuters-Full, all texts are kept as the test set, and in Reuters-Multi, only multi-label texts are kept as the test set. The experimental data description is shown in Table 1.

## Evaluation index

Exact Match Ratio (ER), Micro Averaged Precision (Precision), Micro Averaged Recall (Recall), and Micro Averaged F1 (F1) were used as evaluation indexes in the experiment. Accuracy is used instead of ER in the single-label classification.

## Parameter setting

The experimental parameters of our work are as follows. In the model, input a 300-dimensional word2vec word vector ($d = 300$). The attention module uses two heads of attention ($h = 2$). The first layer of the capsule network module uses 32 convolution filters ($k_1 = 32$), the window size is 3 ($N = 3$). The second layer uses 32 transformation matrices ($k_2 = 32$) and 16-dimensional capsule vectors ($l = 16$). The third layer uses 16 conversion matrices ($k_3 = 16$), the window size is 3 ($N_1 = 3$).The last layer uses 9 capsule vectors ($j = 9$) to represent 9 classes.

In model training, mini-batch with a size of 25 ($batch\_size$) are used, the training batch is controlled to 20 ($Epoch = 20$), and the learning rate is set to 0.001 ($learning\_rate = 0.001$). During the model test, for the single label classification task, the category label corresponding to the capsule vector with the largest module length is taken. For the multi-label classification tasks, the category labels corresponding to capsule vectors with a modulus length greater than 0.5 are taken.

## Benchmark model

In this paper, TextCNN, Capsule-A, and AT-CapsNet are used as benchmark models for comparative experiments. TextCNN is a classic model of text classification based on CNN, which is representative. Capsule-A is a text classification model based on capsule networks. AT-CapsNet is a multi-headed attention capsule network text classification model.

**Table 2 The experimental result of single label classification.** Our model is compared with the benchmark model on three single-label datasets. The evaluation index is accuracy.

| Evaluation index | Accuracy (%) | Precision (%) | Recall (%) | F1 (%) |
|---|---|---|---|---|
| Data set | MR | | | |
| TextCNN | 73.0 | 92.3 | 87.6 | 89.9 |
| Capsule-A | 74.0 | 90.2 | 89.8 | 90.0 |
| AT-CapsNet | 74.8 | 91.5 | 90.0 | 90.7 |
| Syntax-AT-Capsule(ours) | 75.3 | 92.0 | 90.8 | 91.4 |
| Data set | Subj | | | |
| TextCNN | 81.6 | 96.2 | 88.5 | 92.2 |
| Capsule-A | 79.5 | 92.6 | 90.2 | 91.4 |
| AT-CapsNet | 82.4 | 92.2 | 91.0 | 91.6 |
| Syntax-AT-Capsule(ours) | 82.9 | 93.9 | 92.6 | 93.2 |
| Data set | CR | | | |
| TextCNN | 89.5 | 98.2 | 90.6 | 94.2 |
| Capsule-A | 88.7 | 94.0 | 92.0 | 93.0 |
| AT-CapsNet | 89.9 | 96.8 | 93.2 | 95.0 |
| Syntax-AT-Capsule(ours) | 90.6 | 97.3 | 95.3 | 96.3 |

**Table 3 The experimental result of multi label classification.** Our model is compared with the benchmark model on two multi-label datasets. The evaluation index is ER, Precision, Recall and F1.

| Evaluation index | ER (%) | Precision (%) | Recall (%) | F1 (%) |
|---|---|---|---|---|
| Data set | Reuters-Full | | | |
| TextCNN | 85.0 | 97.0 | 86.5 | 91.4 |
| Capsule-A | 84.5 | 92.5 | 90.2 | 91.9 |
| AT-CapsNet | 86.6 | 94.9 | 90.6 | 92.7 |
| Syntax-AT-CapsNet(ours) | 88.1 | 95.0 | 91.8 | 93.4 |
| Data set | Reuters-Multi | | | |
| TextCNN | 28.7 | 96.0 | 59.1 | 73.2 |
| Capsule-A | 44.3 | 86.3 | 75.6 | 80.6 |
| AT-CapsNet | 64.3 | 88.5 | 84.6 | 86.5 |
| Syntax-AT-CapsNet(ours) | 70.3 | 95.2 | 84.9 | 89.7 |

# EXPERIMENTAL RESULTS AND DISCUSSION

## Performance on single-classification and multi-classification tasks

The experimental result of single-label classification is shown in Table 2, and the multi-label classification experiment result is shown in Table 3.

It can be observed from the experimental results:

- Compared with the benchmark model (Table 2), our model achieved the best results of accuracy, Recall, and F1 on the three binary classification data sets ER, Subj, CR. Compared with AT-CapsNet (a multi-headed attention capsule network that does not

**Table 4 The experimental result of Syntax module verification.** Our syntactic module was added to the benchmark model and experimented on Reuters-Full datasets. The evaluation index is ER, Precision, Recall and F1. In each control group, the first model has no syn.

| Data set | Reuters-Full | | | |
|---|---|---|---|---|
| Evaluation index | ER (%) | Precision (%) | Recall (%) | F1 (%) |
| TextCNN | 85.0 | 97.0 | 86.5 | 91.4 |
| Syntax-CNN | 85.9 | 97.1 | 87.7 | 92.1 |
| Capsule-A | 84.5 | 92.5 | 90.2 | 91.9 |
| Syntax-CapsNet | 85.4 | 93.1 | 91.7 | 92.4 |
| AT-CapsNet | 86.6 | 94.9 | 90.6 | 92.7 |
| Syntax-AT-CapsNet(ours) | 88.1 | 95.0 | 91.8 | 93.4 |

introduce syntax), it is found that the three data sets have a significant improvement, which proves the value of introducing syntactic information in this article.

- Compared with the benchmark models (Table 3), on the two multi-label data sets Reuters-Full and Reuters-Multi, our model has achieved competitive results in four evaluation indicators. And achieves the best results in ER, Recall, and F1, which indicates the effectiveness of the model in the classification of multi-tag texts.

That is, our model has better effects on multi-label and single-label classification tasks than benchmark methods. As described in the introduction, the problems with the baseline model are the key to our research. This shows that our model overcomes the shortcomings of these models to a certain extent. That's the purpose of our work.

## Syntax module verification experiment

To show the effect of the syntax module, we did the following experiments and the experimental results are shown in Table 4.

We can see that when the syntactic module is added to the benchmark model, the four evaluation indicators have been significantly improved, which shows that the syntactic module of this article can effectively improve the effect of text classification tasks. It also proves the feasibility and value of extracting syntactic information with graph convolutional neural networks. This also shows that we are right to learn from human reading behavior.

## Module ablation experiment

The results of the module ablation experiment are shown in Tables 5 and 6.

From the above experimental results, we can draw the following conclusions:

- When controlling a single module (Table 5), the ablation of each module will cause the classification effect to decrease to varying degrees, which shows that each module in our model has a certain role in improving the text classification effect. In addition, by comparing the reduced values, it was found that the capsule network had the greatest influence, followed by attention, and the syntax module was smaller, which

**Table 5 The experimental result of control single module ablation.** Remove one module from our model to verify the effect of the module. There is no corresponding module for the first model in each control group.

| Data set | Reuters-Full | | | |
|---|---|---|---|---|
| Evaluation index | ER (%) | Precision (%) | Recall (%) | F1 (%) |
| With/without syntax module | | | | |
| AT-CapsNet | 86.6 | 94.9 | 90.6 | 92.7 |
| Syntax-AT-CapsNet(ours) | 88.1 | 95.0 | 91.8 | 93.4 |
| Decrease value | 1.5 | 0.1 | 1.2 | 0.7 |
| With/without attention module | | | | |
| Syntax-CapsNet | 85.4 | 93.1 | 91.7 | 92.4 |
| Syntax-AT-CapsNet(ours) | 88.1 | 95.0 | 91.8 | 93.4 |
| Decrease value | 2.7 | 1.9 | 0.1 | 1 |
| With/without capsule network module | | | | |
| Syntax-AT-CNN | 85.2 | 94.0 | 91.0 | 92.5 |
| Syntax-AT-CapsNet(ours) | 88.1 | 95.0 | 91.8 | 93.4 |
| Decrease value | 2.9 | 1 | 0.8 | 0.9 |

**Table 6 The experimental result of control two module ablation.** Remove two module from our model to verify the effect of the module. There is no corresponding module for the first model in each control group.

| Data set | Reuters-Full | | | |
|---|---|---|---|---|
| Evaluation index | ER (%) | Precision (%) | Recall (%) | F1 (%) |
| With/without syntax + attention module | | | | |
| CapsNet | 84.5 | 92.5 | 90.2 | 91.9 |
| Syntax-AT-CapsNet(ours) | 88.1 | 95.0 | 91.8 | 93.4 |
| Decrease value | 3.6 | 2.5 | 1.6 | 1.5 |
| With/without syntax + capsule network module | | | | |
| AT-CNN | 84.0 | 94.3 | 87.5 | 90.8 |
| Syntax-AT-CapsNet(ours) | 88.1 | 95.0 | 91.8 | 93.4 |
| Decrease value | 4.1 | 0.7 | 4.3 | 2.6 |
| With/Without Attention + Capsule Network Module | | | | |
| Syntax-CNN | 85.9 | 97.1 | 87.7 | 92.1 |
| Syntax-AT-CapsNet(ours) | 88.1 | 95.0 | 91.8 | 93.4 |
| Decrease value | 2.2 | −2.1 | 4.1 | 1.3 |

indicated the correctness and value of taking the capsule network as the core module in this paper.

- When two modules are controlled (Table 6), the ablation of any two modules will cause the classification effect to decrease to varying degrees. Among them, the ablation syntax and capsule network modules have the greatest influence, and the ablation syntax and attention module have the second influence, and the ablation attention and the

capsule network module has the least influence, which shows that the syntactic module can function to the greatest extent when combined with other modules. It also shows that the motivation of using graph neural networks to encode syntactic information and other models is correct.

- It can be seen from the above that when the attention module, the syntax module, or the capsule network module in the model of this article is removed or partially removed, the effectiveness of the model has declined to vary degrees. Since the syntactic module uses graph convolutional neural networks, the above experiments also prove that graph convolutional neural networks, capsule networks, and multi-head attention have an integrated effect on text classification tasks. This also shows that we are on the right track in building the model.

## CONCLUSIONS

This paper proposes an enhanced capsule network text classification model Syntax-AT-CapsNet for text classification tasks. The model first uses graph convolutional neural networks as submodules to encode syntactic dependency trees, extract syntactic information in text, and further integrate with sequence information and dependency relationships, thereby improving the effect of text classification. Through model classification effect verification experiment, syntax module verification experiment, and module ablation experiment, the effect of the model in this paper on text classification and multi-label text classification task is verified, the function of syntax module is demonstrated, and the integrated effect of graph convolutional neural network, capsule network, and multi-head attention is proved. Future work will further optimize the model for other downstream tasks of text classification.

### Funding
This work was supported by the National Natural Science Foundation of China (No. 61872260). The funders had no role in study design, data collection and analysis, decision to publish, or preparation of the manuscript.

### Grant Disclosures
The following grant information was disclosed by the authors:
National Natural Science Foundation of China: 61872260.

### Competing Interests
The authors declare that they have no competing interests.

### Author Contributions
- Xudong Jia conceived and designed the experiments, performed the experiments, analyzed the data, performed the computation work, prepared figures and/or tables, authored or reviewed drafts of the paper, and approved the final draft.

- Li Wang conceived and designed the experiments, performed the experiments, analyzed the data, performed the computation work, prepared figures and/or tables, authored or reviewed drafts of the paper, and approved the final draft.

## Data Availability

The data is available at GitHub: https://github.com/Mr-jxd/Syntax-AT-CapsNet.

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
