# Peer review of "Attention enhanced capsule network for text classification by encoding syntactic dependency trees with graph convolutional neural network"

_PeerJ Computer Science, doi:10.7717/peerj-cs.831_

## Round 0.1 · original submission · Minor Revisions

Although all three reviewers suggest accepting this manuscript, some minor issues still exist. The technical writing needs to be improved for readability enhancement. For the background and motivation, more details should be introduced for 'related work'. Also, one reviewer noticed table 6 contains a misleading statement, further explanation is helpful in the revised version. Please consider all the comments and re-submit the updated version with careful proofreading.

·

Basic reporting

This paper makes a focused study on the text classification problem from the text syntactic relationship, sequence structure, and semantics. Specifically, the authors first comprehensively summarize the shortcomings of existing models and conclude the challenges for text classification. Later, the author introduced GCN to extract syntactic information for dependencies relationship representation on the one hand and built multi-head attention to encoding the different influence of words to enhance the effect of capsule networks on text classification on the other hand. In the end, the author proved that CapsNet, GCN, and multi-head attention have an integration effect for text classification through experiments.

In general, the author states the research problems very clearly, and presents a very interesting and effective model named Syntax-AT-CapsNet. The paper is organized logically. Thus, I strongly suggest this work should be accepted.

Experimental design

The experiment designed by the author is very sufficient, and the effectiveness of the model (Syntax-AT-CapsNet) proposed by the author can be well proved through the experiment. However, the evaluation metrics could be further enhanced. More diverse evaluation indicators could be added to evaluate the effectiveness of the model in a single-label classification task.

Validity of the findings

The author verified the validity of the conclusion through theoretical description and experimental verification, and it has strong persuasiveness and credibility.

Additional comments

Firstly, several typos should be corrected, such as:
1. At lines 160-161, “... (BERT)also also takes multi-head attention as its basic component.” -> (also) “... (BERT) also takes multi-head attention as its basic component.”
2. At lines 281, “... denotes that Nword vectors in the sentence are connected in series...”-> (N-word) “... denotes that N-word vectors in the sentence are connected in series...”.

In addition, it is suggested that Fig.4 and Fig.6 should be more clear. Maybe it would be better to increase the resolution of the picture.

Finally, the format of the formula in the paper should be unified.

Reviewer 2 ·

Basic reporting

The paper is logically organized, the research content is systematic and the structure is reasonable. The paper considers that the text syntactic relationship and word sequence are important and useful for text classification. So it analyzes the sequence structure and syntactic relations of text, combines the syntactic relationship, sequence structure, and semantics for text representation. And proposes a novel model that utilizes GCN for syntactic relationship, multi-head attention for words and corporate them in capsule network for text classification. The experimental results prove the effectiveness of the proposed method.

Experimental design

The problem researched in this paper is clearly defined. The experimental part is related to the research problem, which is designed reasonably. Experiments on single classification and multi-classification tasks, Syntax module and module ablation experiment were carried out in the experimental part. Experiments prove that graph convolutional neural networks, capsule networks, and multi-head attention have an integrated effect on text classification tasks. And the module ablation experiment shows that each module has a certain role in improving the text classification effect. Methods described with sufficient detail and analyzed the results accordingly.

Validity of the findings

The paper analyzes the background and research status of the problem, proposes an attention enhanced capsule network-based text classification model, which is quite innovative. The technical introduction of the model is clear. The analysis of the experimental results is reasonable, and the data charts are clear. The results of the experiment also verify the effectiveness of the proposed model.

Additional comments

1. It is better to briefly introduce the formulas in section 3 and separately add a new section to describe the details of the algorithm by pseudo-code overall.
2. It's better to give some insight into each module. Not just simply put the conclusions of others’ papers in “related work”. For example, what is the characteristics of CapsNet? Why it is better? I really want to see your own understanding with the formulas. The same as to GCN and multi-head attention. The introduction of those modules is not deep enough.
3. Elaborate on the principle of Syntactic Dependency Tree.
4. There are several mistakes about the format. Such as the label “Figure” in section 3.3 paragraph 2, misses number 3; Section 4.1 paragraph 1, the label “Table” misses number 1, section 4.2 (1), the label “Table 33” repeats number 3, etc. 6. In section 3, the authors should add citations of the formulas.
The description of Equation 17 needs to be clearer.
( Alaparthi, and Mishra 2020) proposed a pre-trained model of language representation (BERT)also also takes multi-head attention as its basic component.
"n-grams" should be "N-grams".
5. Why Table 6 can show that the motivation of using graph neural network to encode syntactic information and other models is correct?

Reviewer 3 ·

Basic reporting

This article proposes a new network architecture for the text classification task. In the feature extraction phase, it uses multi-head attention to encode words' embeddings to extract long-range dependency information, uses GCN to encode Syntactic Dependency Tree to extract syntactic info, and subsequently concatenate them by add operation. Then pass it through a CapsNet for classification.

Experimental design

Generally, the motivation is clear and the experimental work is solid and comprehensive.

Validity of the findings

It's a combination of existed works but results in better performance than some of the previous works.

Additional comments

Follows are some comments to further improve the paper:
1. (1) The English writing of the thesis needs to be further improved and the tenses be unified. Some contents are in the past tense and some are in the present tense. (2) There are a lot of errors in the cross-reference of tables and charts in the paper. There are no corresponding labels behind many tables and figures. (3) The writing of formulas also needs to be more standardized. The corresponding "," needs to be added after most formulas, and where should be lowercase without uppercase. (4) The data set used needs to be added to the corresponding reference instead of a footnote.
2. There are many kinds of methods that can extract syntactic information. Why do the authors choose the GCN instead of other models? I mean what is the benefit of GCN?
3. In addition to the experimental comparison with several baselines, it is also necessary to add the experimental results compared with some other existing algorithms.

---

## Round 0.2 · accepted · Accept

Based on the contribution of the revised version, I would like to recommend accepting this paper as it is.

Reviewer 3 ·

Basic reporting

The authors have addressed all my concerns and thus I think it can be accepted.

Experimental design

The authors have addressed all my concerns and thus I think it can be accepted.

Validity of the findings

The authors have addressed all my concerns and thus I think it can be accepted.